# Validating controlled attenuation parameter in the assessment of hepatic steatosis in living liver donors

Dieter Broering[1], Mohamed Shawkat[1,2], Ali Albenmousa[1], Faisal Abaalkhail[1], Saleh Alabbad[1], Waleed Al-Hamoudi[1], Saad Alghamdi[1], Saleh Alqahthani[1], Ahmad Jaafari[1], Roberto Troisi[1], Khalid Bzeizi[1]*

1 Department of Liver & Small Bowel Transplantation & Hepatobiliary-Pancreatic Surgery, King Faisal Specialist Hospital & Research Center, Riyadh, Saudi Arabia, 2 Internal Medicine Department, Faculty of Medicine, Minia University, Minia, Egypt

* kbzeizi@kfshrc.edu.sa

## Abstract

### Introduction

Hepatic steatosis (HS) negatively impacts transplant outcomes in living liver donors. To date, liver biopsy is preferred for HS evaluation. This study aims to evaluate the measurement of controlled attenuation parameter (CAP) as a diagnostic tool of HS in living liver donors.

### Methods

Candidates recruited to this study, conducted from April 2016 to February 2020, were potential donors who had undergone transient elastography using Fibroscan® and CAP measurements at liver segments VI and VII, followed by liver biopsy. The HS grades from liver biopsy were classified as S0 (<5%), S1 (5–33%), S2 (33–66%), and S3 (>66%). For CAP, they were S0 ($\leq$218dB/m), S1 (218-249dB/m)), S2 (250-305dB/m)), and S3 (>305dB/m)). The CAP measurements were compared with the liver biopsy results.

### Results

Of the 150 potential donors [male, 73.3%; mean age, 30.0±7.0 years; body mass index (BMI), 24.7±3.5kg/m²], 92 (61.3%) had no or mild HS, while 58 (38.7%) and 10% had moderate to severe HS based on CAP and liver biopsy, respectively. Subjects with moderate to severe HS per CAP were mostly males (0.014), and had higher BMI (p = .006), alanine aminotransferase (ALT) (.001), gamma-glutamyl transferase (.026), and high-density lipoprotein (.008). On multivariate analysis, high ALT (OR, 1.051; 95% CI, 1.016–1.087; p = .004) was a predictor of significant HS. The sensitivity, specificity, positive and negative predictive values of CAP to detect significant HS were 93.3%, 67.4, 24.1%, and 98.9%, respectively.

**Data Availability Statement:** All relevant data are within the paper.

**Funding:** No funding was provided for this study.

**Competing interests:** The authors have declared that no competing interests exist.

**Abbreviations:** ALT, Alanine aminotransferase; AUROC, Receiver operator characteristic; BMI, Body mass index; CAP, Controlled attenuation parameter; CLD, Chronic liver disease; CT, Computed tomography; HbA1c, Glycated hemoglobin; HS, Hepatic steatosis; KFSH&RC, King Faisal Specialist Hospital and Research Centre; LDLT, Living donor liver transplantation; MRI, Magnetic resonance imaging; NAFLD, Nonalcoholic fatty liver disease; OR, Odds ratios; ORA, Office of Research Affairs; PPV and NPV, Positive and negative predictive values; TE, Transient elastography; US, Ultrasonography.

## Conclusion

The high sensitivity and negative predictive values of CAP make it a good screening test to exclude significant HS in potential living liver donors which, in turn, can help avoid unnecessary liver biopsies.

## Introduction

For the appropriate selection of living liver donors, the evaluation of hepatic steatosis (HS) is essential because of its considerable negative impact on perioperative outcomes on donor and recipient [1]. Indeed, living donors with significant HS are at risk of increased blood loss during transection and impaired regeneration, which are potentially life-threatening complications, particularly when associated with a reduced volume of the remnant [2, 3]. Moreover, steatotic grafts in recipients are associated with higher rates of primary graft dysfunction, several postoperative complications, and poor overall graft survival [4]. Thus, accurate assessment of the presence and severity of HS in the living liver donor is fundamental to ensure primarily, the donor's safety, as well as successful liver transplantation [4, 5]. Though liver biopsy is the current gold standard for diagnosis and severity assessment of HS, it carries several limitations [6]. Liver biopsy is an invasive procedure associated with several complications, such as bleeding and infection [7]. Further, the sampling technique is variable and carries a high rate of error, while there is the possibility of having non-univocal interpretations among pathologists [7]. Given the limitations of liver biopsy, several non-invasive imaging techniques have been proposed for assessment of HS in living liver donors, such as ultrasonography (US), computed tomography (CT), and magnetic resonance imaging (MRI) [8]. However, these techniques have their own limitations, and none of them have been proven to possess a higher sensitivity in diagnosing HS than liver biopsy [4].

Measurement of controlled attenuation parameter (CAP) is a promising, noninvasive tool for evaluation of HS [9, 10]. It is a qualitative and quantitative technique that measures the degree of ultrasonic attenuation by the hepatic fat at a frequency of 3.5 MHz. CAP is integrated with transient elastography (TE) in the Fibroscan® device (Echosens, Paris, France) [9]. In a recent meta-analysis of 1297 patients with nonalcoholic fatty liver disease (NAFLD) in 2019, the sensitivity of the CAP was 87%, 85%, and 58% in mild, moderate, and severe cases of HS [11]. CAP also has several other advantages, such as being non-invasive, non-ionizing, quantitative, inexpensive, and easy to perform [9]. The only significant limitation of CAP is its reduced ability to discriminate higher HS scores in individuals with a high body mass index (BMI) [12, 13].

Several reports have shown the role of CAP in the assessment of HS in patients with chronic liver disease (CLD) of various etiologies, and these reports suggest threshold CAP values for determining the presence of HS in CLD patients (e.g., > 5% or > 10% of hepatic fat) [14–17]. There is uncertainty, however, on the value of such a scale in those with perceived healthy liver tissue due to the scarcity of data about the role of CAP in the assessment of HS in potential living liver donors. This study aims to evaluate CAP as a diagnostic tool for the presence and severity assessment of HS in potential living liver donors by correlating it to the standard liver biopsy approach.

## Materials and methods

### Sample and setting

We conducted a prospective, single-center, cohort study in the Liver Transplant Unit at King Faisal Specialist Hospital and Research Centre (KFSH&RC) from April 2016 to February 2020. The study was conducted per the latest version of the Declaration of Helsinki and Good Clinical Practice, the policies and procedures of the Office of Research Affairs (ORA) of the KFSH&RC, and the laws of Saudi Arabia. All of the potential transplant donors included in this study provided written informed consent. Potential donors from vulnerable populations (such as minors, the incarcerated, or subjects with compromised mental capacity) were exempt from this study.

The Institutional Review Board within the ORA of KFSH&RC reviewed and approved the study protocol. All subjects signed informed consent before participation.

All potential living liver donors who fulfilled the donor selection criteria and were sent for assessment of HS were eligible for the study. The subjects were selected in a four-step approach according to the center's policy on donor selection for living liver donation (Table 1). Subjects who refused to undergo a liver biopsy, had severe HS on any radiological modality (US, CT or

**Table 1. Workup and steps for selection of living liver donors.**

| |
|---|
| **Step 0 TELEPHONE INTERVIEW (QUESTIONNAIRE)** |
| Age <50 |
| Compatible Blood group |
| Gender |
| Body mass index <30 |
| Previous surgery |
| Medications |
| Relation to recipient |
| **STEP 1** |
| Bloodwork & urinalysis |
| Psychological/ psychiatric consultation |
| Electrocardiography |
| Chest X-ray |
| Pregnancy test for all females |
| **STEP 2** |
| CT abdominal angiography |
| Pulmonary function testing |
| Echocardiography |
| Mammogram (Females aged >40 years) |
| Vaccinations |
| Hepatology consultation |
| Magnetic resonance cholangiopancreatography |
| **STEP 3** |
| preoperative Anesthesiology clinic |
| Carotid Doppler (age >40 years) |
| Colonoscopy and Gastroscopy (age >50 years) |
| Surgical informed consent |
| **STEP4** |
| Liver biopsy |

MRI), planned for left lateral hepatectomy, had BMI above 30 kg/m$^2$, and those who had unsatisfactory CAP assessment (interquartile range > 30%) were excluded from the study.

All donors included in the study were not previously registered as organ donors and they donated for their first or second degree relatives. Recruitment of living donors in our center require that potential donors are adults (18 years and older), come voluntarily for donation, fulfill the criteria mentioned in Table 1 and pass the medical and surgical clearance. The remaining live size should be at least 32% of their liver and should sign an informed consent that explain all the potential complications of the procedure. Donors who have abnormalities in their pre-transplant work up investigations should be cleared from the respective services. All donors who passed the evaluation process during the study period were approached for participation and those who agreed to participate were recruited if they fulfill inclusion and exclusion criteria.

### Data collection

All patients recruited to this study had TE followed by liver biopsy. The TE was performed with Fibroscan® Touch 502 device (Echosens, Paris, France), and CAP measurements were taken at a frequency of 3.5 MHz and a depth of 25–65 mm at liver segments VI and VII [16]. At least ten valid measurements were taken, and the median values were considered for the analysis. Liver biopsies were done under ultrasound guidance from the same liver segments with a full-core biopsy needle (length 33 mm, diameter 18G). The specimens collected were fixed with formalin, embedded in paraffin, stained with hematoxylin and eosin and chromotrope aniline blue stains, and evaluated by an experienced pathologist who was blinded to the result of CAP. The scores used to grade HS on liver biopsy were S0 (<5%), S1 (5%–33%), S2 (33%–66%), and S3 >66%, and the HS evaluation scores for CAP were graded as S0 (≤218 dB/m), S1 (218–249), S2 (250–305), and S3 (>305) [14].

Although the range in S1 is quite wide, however it remains the well-known grading system for reporting severity of steatosis. The degree of steatosis might affect the decision on donation and in most transplant centers subject will be accepted for right lobe donation if they have minimal steatosis (<5% or S0) while steatosis up to 30% (S0 or S1) can be accepted for left lobe donation. Considering subdivision of S1 however will require a separate study to validate such score.

Patient demographics, BMI, and lab investigations [glycated hemoglobin (HbA1c), liver function tests, and lipid profile] at the time of recruitment were collected.

Statistical analysis was performed using the Statistical Package for Social Sciences (SPSS) software version 22.0 (SPSS, Inc., Chicago, IL). Youden index was used to choose the cut-off value for CAP. Quantitative data were expressed as mean and standard deviation. Differences in the groups were analyzed with the unpaired student's t-test for comparison of the means of numerical variables and the chi-squared test for comparison of categorical variables. Multiple logistic regression analysis was used to assess the predictors of HS. P-value < .05 was considered statistically significant. Odds ratios (OR), sensitivity, specificity, positive and negative predictive values (PPV and NPV), positive and negative likelihood ratios, and area under the receiver operator characteristic (AUROC) curve are reported.

### Results

In total, 150 potential living liver donors were included in the analysis. The mean age of the cohort was 30.0±7.0 years, and 73.3% were males, giving a male to female ratio of 3:1. The baseline characteristics of the study subjects are shown in Table 2. Ninety-two subjects (61.3%)

**Table 2. Baseline characteristics of the study population (n = 150).**

| Variable (units) | Values* |
|---|---|
| Age (years)* | 30.0 ± 7.0 |
| Gender (Male/Female) | 110/40 (73.3%/26.7%) |
| BMI (kg/m$^2$)* | 24.7 ± 3.5 |
| HbA1c (%)* | 5.2± 0.5 |
| TB (µmol/L)* | 10.4 ± 6.7 |
| Alb (g/L)* | 47.1± 3.0 |
| ALT (IU/L)* | 24.3 ± 13.1 |
| AST (IU/L)* | 21.3 ± 9.0 |
| ALP (IU/L)* | 72.8 ± 23.1 |
| GGT (IU/L)* | 24.8 ± 0.9 |
| TC (mmol/L)* | 4.4 ± 0.9 |
| TG (mmol/L)* | 1.0 ± 0.9 |
| LDL (mmol/L)* | 3.0 ± 0.8 |
| HDL (mmol/L)* | 1.3 ± 0.3 |

* means ± SD

BMI = body mass index; HbA1C = hemoglobin A1C; TB = total bilirubin; Alb = albumin, ALT = alanine aminotransferase; AST = aspartate aminotransferase; ALP = Alkaline phosphatase; GGT = gamma glutamyl transferase; TC = total cholesterol; TG = triglycerides; LDL = low-density lipoprotein' HDL = high-density lipoprotein

had no or mild HS based on CAP measurement, while moderate to severe HS was detected in 58 subjects (38.7%) compared to 10% on liver biopsy (Fig 1).

On univariate analysis, subjects with moderate to severe HS based on CAP were predominantly males (p = 0.014) and had higher BMI (p = .006), higher HbA1C percentage (p = .04), alanine aminotransferase (ALT) (p = .001), gamma-glutamyl transferase (p = .026) and high-density lipoprotein (p = .008) levels. There was no significant difference in age, aspartate

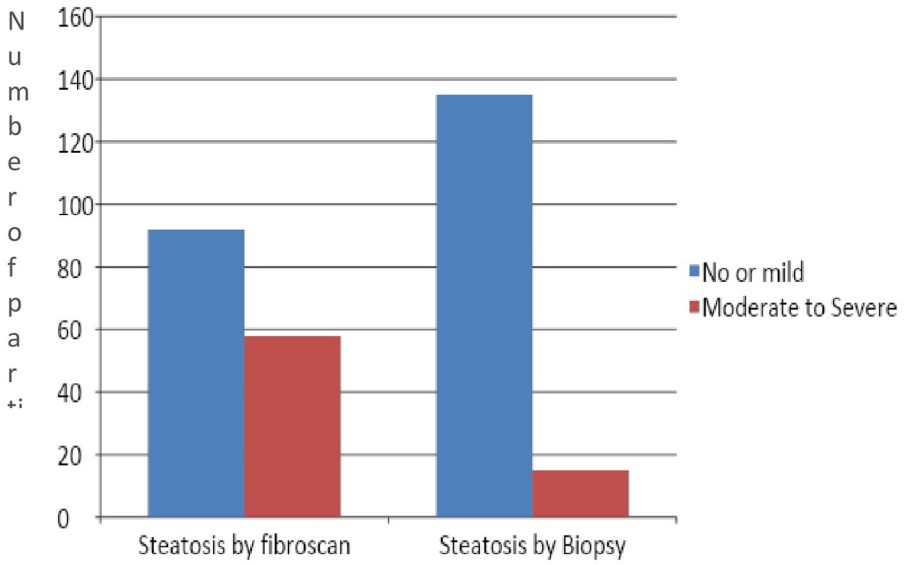

**Fig 1. Steatosis as detected by Fibroscan and liver biopsy.**

**Table 3. Comparison of patients based on the grade of steatosis (ST) by CAP.**

| Variable | No to Mild HS N = 92 | Moderate to Severe HS N = 58 | P-Value |
|---|---|---|---|
| Age | 29.8 ± 6.8 | 30.3 ± 7.3 | 0.671 |
| Gender | 61/31 | 48/9 | 0.014 |
| BMI | 24.1 ± 3.6 | 25.7 ± 3.1 | 0.006 |
| HbA1c | 5.1 ± 0.5 | 5.3 ± 0.4 | 0.04 |
| ALT | 21.0 ± 10.1 | 29.5 ± 15.6 | 0.001 |
| AST | 20.3 ± 8.5 | 22.9 ± 9.6 | 0.089 |
| GGT | 22.5 ± 14.6 | 28.4 ± 17.2 | 0.026 |
| TG | 0.98 ± 1.01 | 1.13 ± 0.67 | 0.324 |
| LDL | 2.9 ± 0.8 | 3.0 ± 1.0 | 0.559 |
| HDL | 1.4 ± 0.4 | 1.2 ± 0.3 | 0.008 |

Results are given as means ± SD

aminotransferase, triglycerides, or low-density lipoprotein levels between groups (Table 3). Multiple logistic regression showed that higher ALT [OR, 1.051; 95% confidence interval (CI), 1.016–1.087; p = .004) was a predictor for moderate to severe HS (Table 4).

The sensitivity, specificity, PPV, and NPV of Fibroscan® to detect significant HS were 93.3%, 67.4%, 24.1%, and 98.9%, respectively. The positive likelihood ratio of CAP was 2.86, and the negative likelihood ratio was 0.1. Assuming that the pre-test probability of having HS (as indicated from biopsy results) was 10%, using the Fagan nomogram, the post-test probability for a positive test was 30%, and 1% for a negative test.

The AUROC curve for CAP was 0.841 (95% CI, 0.736–0.946) when the cut-off for HS used was greater than or equal to a Fibroscan® score of 2 (Fig 2). When the absolute CAP score was used, the AUROC curve was 0.883 (95% CI, 0.725–0.951). Using a cut-off of 276 dB/m, the sensitivity for CAP to detect HS was 86.7%, and specificity was 83.7% (Fig 3).

## Discussion

There is no current consensus regarding how to assess HS in potential living liver donors. Noninvasive scoring methods have been investigated as potential alternatives to liver biopsy, the current gold standard for HS assessment. Due to its potential complications (bleeding, hematomas, or infections), liver biopsy is not universally utilized for assessing the donor liver in living donor liver transplantation (LDLT) [18]. However, having reliable preoperative information on the degree of HS would help to avoid marginal donor grafts, which potentially harm the donors and put recipients at risk of graft dysfunction.

**Table 4. Multiple logistic regression analysis of variables predicting moderate to severe steatosis on CAP.**

| Variable | OR | P value | 95% CI |
|---|---|---|---|
| Ageact | 0.976 | 0.423 | 0.919–1.036 |
| Gender | 0.541 | 0.296 | 0.171–1.711 |
| BMI | 1.134 | 0.058 | 0.996–1.292 |
| ALT | 1.051 | 0.004 | 1.016–1.087 |
| GGT | 0.987 | 0.374 | 0.959–1.016 |
| HDL | 0.522 | 0.369 | 0.126–2.158 |
| HbA1c | 1.764 | 0.203 | 0.736–4.229 |

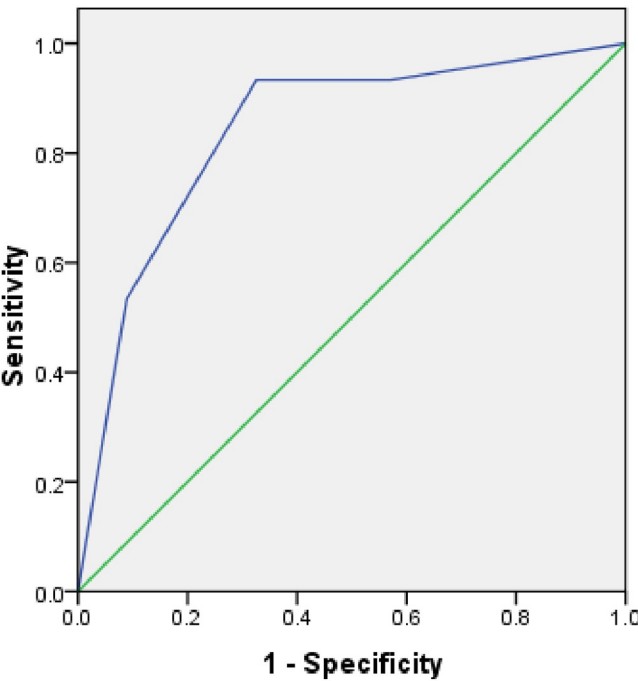

**Fig 2. Receiver operator characteristic (ROC) curve for Fibroscan detection of steatosis for stage >2.**

Ayvazoglu et al. evaluated the importance of preoperative liver biopsy in selecting donor candidates, and they reported that 32% of liver biopsies had pathologic findings. Out of those findings, 44% were fatty changes, with 12% having portal inflammation. In this case, the high rate of pathologic findings in liver biopsy of healthy-appearing donor candidates appeared to indicate the importance of liver biopsy in the preoperative evaluation of donors [19].

To date, only a limited number of studies have reported the potential role of CAP in assessing HS in potential living liver donors [20–22]. This study has shown that CAP has high sensitivity (over 93%) to detect moderate to severe HS and a very high NPV of almost 99%, which can accurately identify patients without significant HS. A score of ≥2 was selected as the best cut-off value using the Youden index. The PPV, however, was low at 24.1%, indicating the need for liver biopsy in donors with positive CAP results to confirm the findings.

Several studies support the use of CAP instead of liver biopsy for HS assessment in various liver diseases. One systematic review evaluated CAP for the diagnosis of HS in patients with various liver conditions; in that study, CAP had good diagnostic accuracy in the evaluation of patients with HS, with similar results regardless of stage [23]. Shi et al. compared the accuracy of CAP versus biopsy, and determined that CAP was accurate for all HS stages [24].

In a study of 224 patients, Baumeler et al. evaluated CAP as a diagnostic tool for identifying the presence and degree of HS in consecutive patients in an outpatient liver unit of a tertiary center [10]. They found that irrespective of the underlying liver disease, CAP values strongly correlated with the degree of HS. These findings support integrating CAP measurements in

## ROC Curve

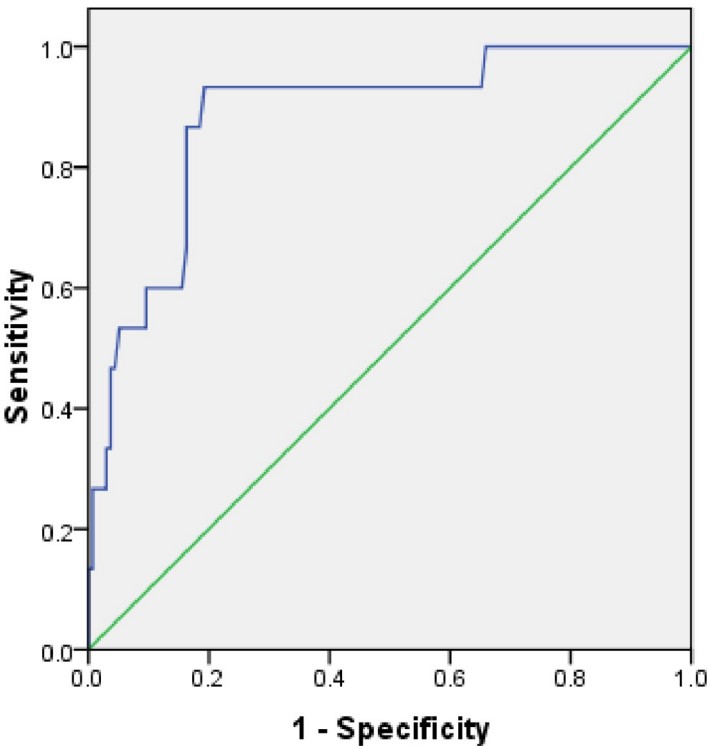

Diagonal segments are produced by ties.

**Fig 3. ROC curve for steatosis using a cutoff of 276 dB/m.**

the standard workup of CLD to identify patients with NAFLD, whether alone or with other causes.

Another systematic review, involving 1297 patients, evaluated the performance of CAP in the diagnosis and staging of HS and biopsy-proven NAFLD [11]. The pooled sensitivity and specificity of CAP in detecting mild HS was 87% and 91%, 85% and 74% for moderate HS, and 76% and 58% for severe HS, respectively. The mean AUROC value for CAP in the diagnosis of mild, moderate, and severe HS was 0.96, 0.82, and 0.70, respectively. Subgroup analysis indicated that variation in the geographic regions, cut-offs, age, and BMI could be the potential sources of heterogeneity in the diagnosis of moderate to severe HS. The study's conclusion notes that CAP should be viewed, albeit with caution, as a potentially sound, noninvasive alternative to liver biopsy.

The findings in this study were well aligned with the results of previous smaller studies using CAP to assess and quantify HS in LDLT. A study of 54 donors by Yen et al., who underwent both CAP and biopsy, also showed high AUROC using a cut-off value of 257 dB/m, 100% sensitivity, and 100% NPV for CAP in evaluating HS. The study found a specific correlation between BMI and CAP values in subjects who did not have HS. They concluded that CAP could prove to be a useful method in identifying HS in living liver donors in East Asia [25].

Hong et al. evaluated the accuracy of CAP for detecting HS in potential liver donors with similar results of high AUROC, albeit with somewhat lower sensitivity and NPV. The CAP

value again correlated positively with BMI, along with waist circumference, hip circumference, magnetic resonance fat signal fraction, and histologic HS grade. This study suggests that CAP, as a simple and noninvasive preoperative assessment for HS, may be sufficient for identifying and thus excluding significant HS in potential living liver donors [26].

Factor affecting the concordance between histology and CAP in assessment of HS like the sampling technique of the biopsy and the adequacy of the tissue are important to consider together with the presence of other liver pathology that might increase the stiffness like liver congestion and infiltration and these are less likely here as we are dealing with healthy subjects.

The findings of our study should be viewed in light of both their strengths and limitations. The cohort was prospectively recruited, and participants underwent a standardized, validated liver disease assessment. The presence and degree of HS were assessed by CAP against the accepted gold standard test (liver biopsy). The study was conducted at a single site, which may limit the generalizability of the results and conclusions. However, the consistency with other studies' findings may imply more ubiquity to these results.

## Conclusion

The high sensitivity and NPV of CAP make it a good screening test for assessment of HS in potential living liver donors to avoid unnecessary liver biopsies in those with a negative result. A positive CAP, however, has poor PPV, and liver biopsy is required to confirm the presence of significant steatosis and exclude other liver pathology.

## Author Contributions

**Conceptualization:** Dieter Broering, Faisal Abaalkhail, Saleh Alabbad, Khalid Bzeizi.

**Data curation:** Mohamed Shawkat, Ali Albenmousa, Waleed Al-Hamoudi, Ahmad Jaafari.

**Formal analysis:** Ali Albenmousa.

**Methodology:** Ali Albenmousa.

**Project administration:** Dieter Broering, Mohamed Shawkat.

**Supervision:** Dieter Broering, Khalid Bzeizi.

**Validation:** Ali Albenmousa, Faisal Abaalkhail, Saad Alghamdi, Saleh Alqahthani, Roberto Troisi.

**Writing – original draft:** Khalid Bzeizi.

**Writing – review & editing:** Dieter Broering, Saleh Alabbad, Waleed Al-Hamoudi, Saad Alghamdi, Saleh Alqahthani, Roberto Troisi, Khalid Bzeizi.

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
