## [Decision Letter · Decision Letter 0]

24 Nov 2020

PONE-D-20-31215

VALIDATING CONTROLED ATTENUATION PARAMETER IN THE ASSESSMENT OF HEPATIC STEATOSIS IN LIVING LIVER DONORS

PLOS ONE

Dear Dr. Bzeizi,

Thank you for submitting your manuscript to PLOS ONE. After careful consideration, we feel that it has merit but does not fully meet PLOS ONE’s publication criteria as it currently stands. Therefore, we invite you to submit a revised version of the manuscript that addresses the points raised during the review process.

This is an important study for those in the field of living liver donor work up; it describes an innovative way (CAP) to assess liver steatosis in potential donors, which could replace the liver biopsy as gold standard, which would reduce the risk for potential donors, which is something we should always aspire to. The authors should be applauded for this work, and expert reviewers have highlighted its value for the literature. They have recommended minor revisions to be made, and I would encourage the authors to revise the MS accordingly and respond in a point-by-point fashion to the comments. Looking forward to receive the revised MS.

We look forward to receiving your revised manuscript.

Kind regards,

Frank JMF Dor, M.D., Ph.D., FEBS, FRCS

Academic Editor

PLOS ONE

Journal Requirements:

2. Thank you for stating in the text of your manuscript "The Institutional Review Board within the ORA of KFSH&RC reviewed and approved the study protocol. All subjects signed informed consent before participation." Please also add this information to your ethics statement in the online submission form.

3. In your Methods section, please provide additional information about the participant recruitment method and the demographic details of your participants. Please ensure you have provided sufficient details to replicate the analyses such as:

a) a statement as to whether your sample can be considered representative of a larger population, and

b) a description of how participants were recruited.

4. We note that your study involved studying potential tissue/organ transplantation donors. Please provide the following information:

a) Please state in your response letter and ethics statement whether the potential donors involved any vulnerable populations; for example, prisoners, subjects with reduced mental capacity due to illness or age, or minors.

- If a vulnerable population was used, please describe the population, justify the decision to use them as potential organ donors, and clearly describe what measures were taken in the informed consent procedure to assure protection of the vulnerable group and avoid coercion.

- If a vulnerable population was not used, please state in your ethics statement, “None of the potential transplant donors was from a vulnerable population and all potential donors provided written informed consent that was freely given.”

b) Please provide a blank example of the form used to obtain consent from donors, and an English translation if the original is in a different language.

c) Please indicate whether the donors were previously registered as organ donors.

d) Please discuss whether medical costs were covered or other cash payments were provided to the potential donor. If so, please specify the value of this support (in local currency and equivalent to U.S. dollars).

5. Please include additional information regarding the questionnaire used for selecting living donors in the study and ensure that you have provided sufficient details that others could replicate the analyses. For instance, if you developed a questionnaire as part of this study and it is not under a copyright more restrictive than CC-BY, please include a copy, in both the original language and English, as Supporting Information.

6.  Thank you for stating the following financial disclosure:

7. Thank you for submitting the above manuscript to PLOS ONE. During our internal evaluation of the manuscript, we found significant text overlap between your submission and the following previously published works, on some of which you may be an author.

https://www.sciencedirect.com/science/article/pii/S0041134518308984?via%3Dihub

https://smw.ch/article/doi/smw.2019.20077

https://bmcgastroenterol.biomedcentral.com/articles/10.1186/s12876-019-0961-9

https://journals.lww.com/eurojgh/Abstract/2017/07000/Clinical_usefulness_of_controlled_attenuation.10.aspx

Please revise the manuscript to rephrase the duplicated text, cite your sources, and provide details as to how the current manuscript advances on previous work. Please note that further consideration is dependent on the submission of a manuscript that addresses these concerns about the overlap in text with published work.

Reviewers' comments:

Reviewer's Responses to Questions

**Comments to the Author**

1. Is the manuscript technically sound, and do the data support the conclusions?

Reviewer #1: Yes

Reviewer #2: Yes

2. Has the statistical analysis been performed appropriately and rigorously? 

Reviewer #1: Yes

Reviewer #2: Yes

3. Have the authors made all data underlying the findings in their manuscript fully available?

Reviewer #1: Yes

Reviewer #2: Yes

4. Is the manuscript presented in an intelligible fashion and written in standard English?

Reviewer #1: Yes

Reviewer #2: Yes

5. Review Comments to the Author

Reviewer #1: Dear authors,

congrats for your paper describing a non invasive tool to investigate hepatic steatosis in the assessment of living liver donors.

I have two comments:

You're using the following scores to grade hepatic steatosis on liver biopsy - S0 (<5%), S1 (5%–33%), S2 (33%–66%), and S3 >66%. Decision making whether or not a living donor can be accepted and if yes, whether to use a right or left lobe, depends amongst other things on the liver remnant in % and the liver quality, e.g. hepatic steatosis (HS). In this regards score S1 includes a wide range of hepatic steatosis between 5 and 33 %. This range includes also a wide range of possible decisions whether or not a living donor is suitable or not at all, respectively if suitable for a right and left or only a left lobe. It seems that a subdivision of S1 would be valid, but should be at least discussed.

On page 15 is written "In the 54 donors of the study by xx, ...". I guess "xx" is just a spacer for the planned citation.

Reviewer #2: This is a very interesting paper about the potential use of the controlled attenuation parameter (CAP) in the assessment of hepatic steatosis in living liver donors.

The current gold standard is the liver biopsy but CAP may be a useful tool to avoid unnecessary biopsies, although it is still necessary due to the reported low diagnostic power of CAP in presence of severe steatosis,

This is a well-conducted prospective study, abstract, introduction, and discussion are clear and concise, aim and conclusion are coherent. Figures and tables are appropriate for the manuscript.

To-date, data on this specific topic (evaluation of steatosis in perceived healthy liver) are scarce and from relatively small case-series.

Evaluation of the necessary statistical power to use the term "validating" is required.

Other issues include:

- an evaluation of the parameters affecting the concordance between CAP and biopsy could add some value to this paper

- it is reported that an expert pathologist evaluates the biopsy blindly, but there is no specific indication about who performs the CAP, which is an operator-dependent test.

- why exclude left lateral hepatectomies from inclusion criteria?

Minor revisions:

- please revise the language used in the Methods section of the abstract

- the use of the Youden index to choose the cut-off should be mentioned first in the Method section

- "Another systematic review of the performance of CAP ... 1297 patients" is maybe linked to the ref 11 instead of 23

- "In the 54 donors of the study by xx" should be revised

- I would put the paragraph with the reference about the paper of Ayvazoglu at the beginning of the discussion since they stress the importance of the biopsy.

6. PLOS authors have the option to publish the peer review history of their article (what does this mean?). If published, this will include your full peer review and any attached files.

Reviewer #1: **Yes: **Markus U. Boehnert

Reviewer #2: No

---

## [Author Response · Author response to Decision Letter 0]

9 Feb 2021

January 11, 2021

Frank JMF Dor, M.D., Ph.D., FEBS, FRCS

Academic Editor

PLOS ONE

Dear Professor Dor,

Re: “Validating controlled attenuation parameter in the assessment of hepatic steatosis in living liver donors”

Many thanks for giving us the opportunity to resubmit the manuscript after the revision as per your valuable suggestions and those of the reviewers. The manuscript has been proofread by a native English speaker with expertise in the field.

I hope that we covered most of the points raised and once again, many thanks for considering our study for publication in PLOS ONE

Best wishes,

Khalid Bzeizi

Point-By- Point Response to The Comments Raised by The Reviewers:

3. In your Methods section, please provide additional information about the participant recruitment method and the demographic details of your participants. Please ensure you have provided sufficient details to replicate the analyses such as:

a) a statement as to whether your sample can be considered representative of a larger population, and

b) a description of how participants were recruited.

3). Answer:

The preliminary and essential requirement for accepting potential donors to proceed with the liver transplant workup include:

1). Signing an informed consent for donation, (2). Being a direct relative to the patient, (3) Age between 18-50 years, (4). Body mass index (BMI) <30, (5). No previous abdominal surgery, (6). No history of alcohol or substance abuse, (7). No documented chronic medical history or on regular medication and (8). All donors were cleared by psychologist/psychiatrist from any psychiatric illness or of reduced mental capacity 

b). Given the nature of donor process selection, the sample is not representative of the general population.

c). All donors were direct relatives of the patients.

4. We note that your study involved studying potential tissue/organ transplantation donors. Please provide the following information:

a) Please state in your response letter and ethics statement whether the potential donors involved any vulnerable populations; for example, prisoners, subjects with reduced mental capacity due to illness or age, or minors.

Answer: The following statement was added in page 6, para 2:

“None of the potential transplant donors was from a vulnerable population (e.g, prisoners, subjects with reduced mental capacity due to illness or age, or minors) and all potential donors provided written informed consent that was freely given.” 

- If a vulnerable population was used, please describe the population, justify the decision to use them as potential organ donors, and clearly describe what measures were taken in the informed consent procedure to assure protection of the vulnerable group and avoid coercion. 

Answer: NA

- If a vulnerable population was not used, please state in your ethics statement, “None of the potential transplant donors was from a vulnerable population and all potential donors provided written informed consent that was freely given.”

Answer: Done (Page 6, Para2)

b) Please provide a blank example of the form used to obtain consent from donors, and an English translation if the original is in a different language.

Answer: The English consent form has been uploaded

c) Please indicate whether the donors were previously registered as organ donors.

Answer: No. This is unfortunately not the practice in KSA

d) Please discuss whether medical costs were covered or other cash payments were provided to the potential donor. If so, please specify the value of this support (in local currency and equivalent to U.S. dollars).

Answer: None

 5. Please include additional information regarding the questionnaire used for selecting living donors in the study and ensure that you have provided sufficient details that others could replicate the analyses. For instance, if you developed a questionnaire as part of this study and it is not under a copyright more restrictive than CC-BY, please include a copy, in both the original language and English, as Supporting Information.

 6. Thank you for stating the following financial disclosure:

1. Please clarify the sources of funding (financial or material support) for your study. List the grants or organizations that supported your study, including funding received from your institution.

2. State what role the funders took in the study. If the funders had no role in your study, please state: “The funders had no role in study design, data collection and analysis, decision to publish, or preparation of the manuscript.”

3. If any authors received a salary from any of your funders, please state which authors and which funders.

4. If you did not receive any funding for this study, please state: “The authors received no specific funding for this work.”

Answer: No funding was provided for this study. All workup assessment measures were done as part of the transplant protocol requirements 

7. Thank you for submitting the above manuscript to PLOS ONE. During our internal evaluation of the manuscript, we found significant text overlap between your submission and the following previously published works, on some of which you may be an author.

https://www.sciencedirect.com/science/article/pii/S0041134518308984?via%3Dihub

https://smw.ch/article/doi/smw.2019.20077

https://bmcgastroenterol.biomedcentral.com/articles/10.1186/s12876-019-0961-9

https://journals.lww.com/eurojgh/Abstract/2017/07000/Clinical_usefulness_of_controlled_attenuation.10.aspx

Please revise the manuscript to rephrase the duplicated text, cite your sources, and provide details as to how the current manuscript advances on previous work. Please note that further consideration is dependent on the submission of a manuscript that addresses these concerns about the overlap in text with published work.

Answer: Many thanks for the valuable comment. The manuscript has been revised taking into consideration the overlaps mentioned. This has now been rectified

5. Review Comments to the Author

Reviewer #1: Dear authors,

congrats for your paper describing a non invasive tool to investigate hepatic steatosis in the assessment of living liver donors.

I have two comments:

You're using the following scores to grade hepatic steatosis on liver biopsy - S0 (<5%), S1 (5%–33%), S2 (33%–66%), and S3 >66%. Decision making whether or not a living donor can be accepted and if yes, whether to use a right or left lobe, depends amongst other things on the liver remnant in % and the liver quality, e.g. hepatic steatosis (HS). In this regards score S1 includes a wide range of hepatic steatosis between 5 and 33 %. This range includes also a wide range of possible decisions whether or not a living donor is suitable or not at all, respectively if suitable for a right and left or only a left lobe. It seems that a subdivision of S1 would be valid, but should be at least discussed.

Answer: We have addressed this point in the methods section (page 7, para: 2):

Although the range in S1 is quite wide, however it remains the well-known grading system for reporting severity of steatosis. The degree of steatosis might affect the decision on donation and in most transplant centers subject will be accepted for right lobe donation if they have minimal steatosis (<5% or S0) while steatosis up to 30% (S0 or S1) can be accepted for left lobe donation. Considering subdivision of S1 however will require a separate study to validate such score.

On page 15 is written "In the 54 donors of the study by xx, ...". I guess "xx" is just a spacer for the planned citation.

Answer: Thank you. This has been rectified (page 10, para 2):

“In the 54 donors of the study by Yen et al, who underwent both CAP and biopsy, also showed high AUROC using a cut-off value of 257 dB/m, 100% sensitivity, and 100% NPV for CAP in evaluating HS.”

Reviewer #2: This is a very interesting paper about the potential use of the controlled attenuation parameter (CAP) in the assessment of hepatic steatosis in living liver donors.

The current gold standard is the liver biopsy but CAP may be a useful tool to avoid unnecessary biopsies, although it is still necessary due to the reported low diagnostic power of CAP in presence of severe steatosis,

This is a well-conducted prospective study, abstract, introduction, and discussion are clear and concise, aim and conclusion are coherent. Figures and tables are appropriate for the manuscript.

To-date, data on this specific topic (evaluation of steatosis in perceived healthy liver) are scarce and from relatively small case-series.

Evaluation of the necessary statistical power to use the term "validating" is required.

Other issues include:

- an evaluation of the parameters affecting the concordance between CAP and biopsy could add some value to this paper

Answer: Many thanks for the suggestion. This has been added in the discussion (page 11, para:8):

“Factor affecting the concordance between histology and CAP in assessment of HS like the sampling technique of the biopsy and the adequacy of the tissue are important to consider together with the presence of other liver pathology that might increase the stiffness like liver congestion and infiltration and these are less likely here as we are dealing with healthy subjects.”

- it is reported that an expert pathologist evaluates the biopsy blindly, but there is no specific indication about who performs the CAP, which is an operator-dependent test. 

Answer: One operator (Dr M. Shawkat) performed the CAP

- why exclude left lateral hepatectomies from inclusion criteria?

Answer: As part of our center protocol. none of the donors selected for left lateral hepatectomies needed to undergo liver biopsy as a requirement for donation. 

Minor revisions:

- please revise the language used in the Methods section of the abstract.

Answer: Done. 

- the use of the Youden index to choose the cut-off should be mentioned first in the Method section

Answer: Many thanks for this suggestion. This has been added to the manuscript (page:7, para: 4)

“Youden index was used to choose the cut-off value for CAP.”

- "Another systematic review of the performance of CAP ... 1297 patients" is maybe linked to the ref 11 instead of 23.

Answer: Many thanks for highlighting this reference mislabeling. This has now been rectified in the manuscript (page: 9, para: 2).

- "In the 54 donors of the study by xx" should be revised

Answer: Thank you. This has been rectified (page 10, para 2):

“In the 54 donors of the study by Yen et al, who underwent both CAP and biopsy, also showed high AUROC using a cut-off value of 257 dB/m, 100% sensitivity, and 100% NPV for CAP in evaluating HS.”

- I would put the paragraph with the reference about the paper of Ayvazoglu at the beginning of the discussion since they stress the importance of the biopsy.

Answer: Many thanks for the suggestion. The authors have felt that the paragraph describing the findings of Ayvazoglu et al study is reasonably positioned in the discussion section and we prefer to keep it unchanged.

---

## [Decision Letter · Decision Letter 1]

10 Mar 2021

PONE-D-20-31215R1

VALIDATING CONTROLED ATTENUATION PARAMETER IN THE ASSESSMENT OF HEPATIC STEATOSIS IN LIVING LIVER DONORS

PLOS ONE

Dear Dr. Bzeizi,

Thank you for submitting your manuscript to PLOS ONE. After careful consideration, we feel that it has merit but does not fully meet PLOS ONE’s publication criteria as it currently stands. Therefore, we invite you to submit a revised version of the manuscript that addresses the points raised during the review process.

We look forward to receiving your revised manuscript.

Kind regards,

Frank JMF Dor, M.D., Ph.D., FEBS, FRCS

Academic Editor

PLOS ONE

Journal Requirements:

Additional Editor Comments (if provided):

Congratulations on the revised MS. Reviewer 2 suggests a few further edits to be made, and i am confident you can address any remaining issues in the revisions.

Reviewers' comments:

Reviewer's Responses to Questions

**Comments to the Author**

1. If the authors have adequately addressed your comments raised in a previous round of review and you feel that this manuscript is now acceptable for publication, you may indicate that here to bypass the “Comments to the Author” section, enter your conflict of interest statement in the “Confidential to Editor” section, and submit your "Accept" recommendation.

Reviewer #1: All comments have been addressed

Reviewer #2: (No Response)

2. Is the manuscript technically sound, and do the data support the conclusions?

Reviewer #1: (No Response)

Reviewer #2: Yes

3. Has the statistical analysis been performed appropriately and rigorously? 

Reviewer #1: (No Response)

Reviewer #2: N/A

4. Have the authors made all data underlying the findings in their manuscript fully available?

Reviewer #1: (No Response)

Reviewer #2: Yes

5. Is the manuscript presented in an intelligible fashion and written in standard English?

Reviewer #1: (No Response)

Reviewer #2: Yes

6. Review Comments to the Author

Reviewer #1: (No Response)

Reviewer #2: This is a very interesting paper about the potential use of the CAP in the assessment of hepatic steatosis in living liver donors.

Liver biopsy remains the current gold standard and, but CAP, despite the low diagnostic power of CAP with severe steatosis, may be a useful tool to avoid unnecessary biopsies.

This is a well-conducted prospective study. Abstract , introduction, and discussion are clear and concise and aim and conclusion are coherent. Figures and tables are adequate and coherent to the test.

To-date, data on this specific topic (assessment of steatosis in perceived healthy livers) are scarce and from relatively small case-series.

Evaluation of the necessary statistical power to use the term "validating" is required.

Further issues with this manuscript:

- an evaluation of the parameters affecting the concordance between CAP and biopsy could add value to this paper

- it is reported that an expert pathologist assessed the biopsies in a blind way. However it is not clear who performs CAP that is an operator-dependent exam

- why was left lateral hepatectomy chosen among the exclusion criteria?

Minor comments:

- please revise the language used in the methods section of the abstract

- the use of the Youden index to identify the cut-off should be mentioned with priority in the Methods section

- "Another systematic review of the performace of CAP ... 1297 patients" is possibly linked to ref 11 instead of 23 ?

- "In the 54 donors of the study by xx" should be revised

- The paragraph with the reference regarding the paper of Ayvazoglu should find a place earlier in the Discussion given that they clarify the importance of the biopsy.

7. PLOS authors have the option to publish the peer review history of their article (what does this mean?). If published, this will include your full peer review and any attached files.

Reviewer #1: **Yes: **Markus U. Boehnert

Reviewer #2: No

---

## [Author Response · Author response to Decision Letter 1]

11 Apr 2021

Dear Professor Dor,

Ref: PONE-D-20-31215R1

VALIDATING CONTROLED ATTENUATION PARAMETER IN THE ASSESSMENT OF HEPATIC STEATOSIS IN LIVING LIVER DONORS

PLOS ONE

Many thanks for the opportunity to resubmit a revised manuscript and we appreciate the valuable comments of the reviewers.

The following are the responses to the points raised by Reviewer #2:

1) An evaluation of the parameters affecting the concordance between CAP and biopsy could add value to this paper.

Response: In the earlier submission, we alluded to the factors and parameters that might impact on the concordance between CAP and biopsy. In paragraph-1 (page 11), we highlighted the fact that the donors are healthy cohorts and this might well be a factor in minimizing the potential factors that adversely impact on the concordance.

 “Factor affecting the concordance between histology and CAP in assessment of HS like the sampling technique of the biopsy and the adequacy of the tissue are important to consider together with the presence of other liver pathology that might increase the stiffness like liver congestion and infiltration and these are less likely here as we are dealing with healthy subjects.”

2) It is reported that an expert pathologist assessed the biopsies in a blind way. However, it is not clear who performs CAP that is an operator-dependent exam.

Response: The Fibroscan & CAP were performed by a certified technician under the supervision of one hepatologist. The technician did more than 1000 scan on potential donors prior to his participation in this study.

3) Why was left lateral hepatectomy chosen among the exclusion criteria?

Response: As per our department transplant protocol, liver biopsy is not a requirement from donors of left lateral segment. The liver biopsies were only for right lobe donors.

4) Please revise the language used in the methods section of the abstract.

Response: Many thanks for the suggestion. These parts have been proofread.

5) The use of the Youden index to identify the cut-off should be mentioned with priority in the Methods section.

Response: This point is now highlighted in the last paragraph (page 7).

6) "Another systematic review of the performace of CAP ... 1297 patients" is possibly linked to ref 11 instead of 23 ?.

Response: Many thanks for the suggestions. This has now been corrected (para: 3, page 10):

Another systematic review, involving 1297 patients, evaluated the performance of CAP in the diagnosis and staging of HS and biopsy-proven NAFLD.(11)

7) "In the 54 donors of the study by xx" should be revised.

Response: Thank you. This has now been revised (para 4: page 10):

“ A study of 54 donors by Yen et al, who underwent both CAP and biopsy, also showed high AUROC using a cut-off value of 257 dB/m, 100% sensitivity, and 100% NPV for CAP in evaluating HS”.

8) The paragraph with the reference regarding the paper of Ayvazoglu should find a place earlier in the Discussion given that they clarify the importance of the biopsy.

Response: Many thanks for the suggestion. This paragraph has now moved up to page 9 (para: 2). The order of the references has now changed upon the transfer of this paragraph

We hope that the points raised by the reviewers have been addressed and once again, many thanks for the opportunity to resubmit the manuscript.

Best wishes

Khalid Bzeizi

---

## [Decision Letter · Decision Letter 2]

28 Apr 2021

VALIDATING CONTROLED ATTENUATION PARAMETER IN THE ASSESSMENT OF HEPATIC STEATOSIS IN LIVING LIVER DONORS

PONE-D-20-31215R2

Dear Dr. Bzeizi,

We’re pleased to inform you that your manuscript has been judged scientifically suitable for publication and will be formally accepted for publication once it meets all outstanding technical requirements.

Kind regards,

Frank JMF Dor, M.D., Ph.D., FEBS, FRCS

Academic Editor

PLOS ONE

Additional Editor Comments (optional):

Reviewers' comments:

Reviewer's Responses to Questions

**Comments to the Author**

1. If the authors have adequately addressed your comments raised in a previous round of review and you feel that this manuscript is now acceptable for publication, you may indicate that here to bypass the “Comments to the Author” section, enter your conflict of interest statement in the “Confidential to Editor” section, and submit your "Accept" recommendation.

Reviewer #2: All comments have been addressed

2. Is the manuscript technically sound, and do the data support the conclusions?

Reviewer #2: Yes

3. Has the statistical analysis been performed appropriately and rigorously? 

Reviewer #2: Yes

4. Have the authors made all data underlying the findings in their manuscript fully available?

Reviewer #2: Yes

5. Is the manuscript presented in an intelligible fashion and written in standard English?

Reviewer #2: Yes

6. Review Comments to the Author

Reviewer #2: The authors correctly and completely addressed all the issues raised. In my opinion, the topic is of interest for the readership

7. PLOS authors have the option to publish the peer review history of their article (what does this mean?). If published, this will include your full peer review and any attached files.

Reviewer #2: **Yes: **Paolo Muiesan

---

## [Editor Report · Acceptance letter]

4 May 2021

PONE-D-20-31215R2 

Validating controlled attenuation parameter in the assessment of hepatic steatosis in living liver donors 

Dear Dr. Bzeizi:

I'm pleased to inform you that your manuscript has been deemed suitable for publication in PLOS ONE. Congratulations! Your manuscript is now with our production department. 

Kind regards, 

on behalf of

Dr. Frank JMF Dor 

Academic Editor

PLOS ONE